# Application of Virtual Reality Systems in Bone Trauma Procedures

**DOI:** 10.3390/medicina59030562

**Published:** 2023-03-14

**Authors:** Chiedozie Kenneth Ugwoke, Domenico Albano, Nejc Umek, Ivo Dumić-Čule, Žiga Snoj

**Affiliations:** 1Institute of Anatomy, Faculty of Medicine, University of Ljubljana, Korytkova Ulica 2, 1000 Ljubljana, Slovenia; nejc.umek@mf.uni-lj.si; 2Unità Operativa di Radiologia Diagnostica ed Interventistica, IRCCS Istituto Ortopedico Galeazzi, Via Riccardo Galeazzi 4, 20161 Milano, Italy; 3Department of Nursing, University North, 104. Brigade 3, 42000 Varaždin, Croatia; 4Department of Diagnostic and Interventional Radiology, University Hospital Center Zagreb, Kišpatićeva Ulica 12, 10000 Zagreb, Croatia; 5Department of Radiology, Faculty of Medicine, University of Ljubljana, Vrazov trg 2, 1000 Ljubljana, Slovenia; 6Clinical Institute of Radiology, University Medical Centre Ljubljana, Zaloška 7, 1000 Ljubljana, Slovenia

**Keywords:** virtual reality, simulation, fracture, bone trauma, orthopaedics, preoperative planning

## Abstract

*Background and Objectives*: Bone fractures contribute significantly to the global disease and disability burden and are associated with a high and escalating incidence and tremendous economic consequences. The increasingly challenging climate of orthopaedic training and practice re-echoes the established potential of leveraging computer-based reality technologies to support patient-specific simulations for procedural teaching and surgical precision. Unfortunately, despite the recognised potential of virtual reality technologies in orthopaedic surgery, its adoption and integration, particularly in fracture procedures, have lagged behind other surgical specialities. We aimed to review the available virtual reality systems adapted for orthopaedic trauma procedures. *Materials and Methods*: We performed an extensive literature search in Medline (PubMed), Science Direct, SpringerLink, and Google Scholar and presented a narrative synthesis of the state of the art on virtual reality systems for bone trauma procedures. *Results*: We categorised existing simulation modalities into those for fracture fixation techniques, drilling procedures, and prosthetic design and implantation and described the important technical features, as well as their clinical validity and applications. *Conclusions*: Over the past decade, an increasing number of high- and low-fidelity virtual reality systems for bone trauma procedures have been introduced, demonstrating important benefits with regard to improving procedural teaching and learning, preoperative planning and rehearsal, intraoperative precision and efficiency, and postoperative outcomes. However, further technical developments in line with industry benchmarks and metrics are needed in addition to more standardised and rigorous clinical validation.

## 1. Introduction

### 1.1. The Burden of Bone Trauma

Traumatic bone conditions are an important contributor to global disease and disability burden. A worldwide incidence estimate using the framework of the Global Burden of Diseases, Injuries, and Risk Factors Study showed that there were 178 million new fractures in 2019, representing an increase of 33.4% since 1990 [1]. At 11.58%, the global incidence of imminent fracture is also worrisome [2]. In the United States, about 340,000 hip fractures are reported annually among elderly patients, and across Europe, about 600,000 hip fractures were reported in 2010 [3,4,5,6]. The incidence of musculoskeletal injuries in low- and middle-income countries is estimated to be between 779 and 1574 per 100,000 person years [7]. In developed countries with an ageing population, the bulk of the fracture burden is related to degenerative bone conditions such as osteoporosis, while in developing countries, traumatic injuries such as road traffic accidents account for the vast majority of fractures [5,7]. Although the COVID-19 pandemic has contributed to a global decline in the overall fracture burden, the rate of fragility fractures remained unchanged [8,9,10]. Refractures occur in approximately 50% of index fragility fractures within two years [2], and in the United States, post-traumatic osteoarthritis accounts for about 12% of symptomatic osteoarthritis [11]. Traumatic orthopaedic conditions are further associated with a considerable economic burden. In 2017, the annual cost of fragility fractures was estimated at EUR 37.5 billion in the five largest European Union states plus Sweden, with an additional loss of 1.0 million quality-adjusted life years [12].

### 1.2. Rationale and Fundamentals of Virtual Reality in Bone Trauma

Coexisting with the escalating burden of traumatic bone conditions is an increasingly challenging climate of orthopaedic training and practice. The changing trends in clinical work hours, healthcare budgets, legislative frameworks, patient safety considerations, and public expectations significantly impact surgical training and practice in the current era [13,14,15]. A worrisome decline in the volume of orthopaedic and trauma specialists has been well-documented over the past few decades [16,17,18,19]. Compounding the existing workforce shortages, the disturbances to global healthcare systems due to the COVID-19 pandemic caused further negative disruptions to orthopaedic and trauma care and education [20,21,22,23]. These realities have further re-echoed the established potential of leveraging computer-based technologies to enhance the efficiency of orthopaedic training and practice. Among other benefits, the diverse developments in computer-assisted surgery and simulation technologies are expected to foster improvements in surgical precision, safety, and outcomes [24,25,26,27].

Driven by the rapid evolution of computing power and imaging capabilities over the past few decades, several reality technologies have been developed for clinical and pedagogical applications. In orthopaedic trauma surgery, different modalities of reality technologies are currently available to support procedural teaching, practice, and patient-specific simulations, namely, virtual reality (VR), augmented reality, and mixed reality. These three reality modalities may sometimes be collectively described with the umbrella term *extended reality* [28] (Figure 1). VR systems provide total visual immersion, movements, and interactions in an artificial, computer-generated environment that may incorporate artificial stimuli such as sounds or hand-operated controllers to improve the interactive experience [29]. VR systems increasingly combine haptic feedback to simulate touch, vibration, and motion, in addition to standard components such as a 3D-capable computer, head-mounted display, and controllers with position sensors [25]. While VR employs an entirely artificial computer-generated environment, augmented reality superimposes digital images onto physical environments in real-time. The term *augmented virtual reality* is also used distinctively in some literature to describe the real-time representation of physical world events in a virtual environment [28]. On the other hand, mixed reality utilises a digital display overlay in conjunction with interactive projected holograms, allowing the operator to explore the physical environment while simultaneously interacting with and controlling the digital material provided by the device. [29,30]. Table 1 summarises the important features of virtual, augmented, and mixed reality surgical systems.

The arguments for integrating VR technologies into orthopaedic practice are compelling, especially considering the widening spectrum of technically challenging procedures in the field. Unfortunately, despite the recognised potential, the adoption of VR and other reality technologies in orthopaedics has lagged behind other surgical specialities [31,32,33]. While there have been notable advancements in the integration of VR for arthroscopic surgeries [34,35,36,37], the evolution of VR technologies for bone fracture procedures has followed a more modest course [31]. The present review aims to describe the state of the art on VR systems for orthopaedic trauma procedures.

## 2. Methodology

### 2.1. Search Strategy

An extensive literature search was performed on Medline (PubMed), Science Direct, SpringerLink, and Google Scholar databases to identify original reports and relevant reviews focusing on the review subject. The following medical subject headings (MeSH) terms were used in various combinations using the Boolean operators “AND” and “OR” in accordance with the advanced search algorithms of the searched databases: “virtual reality” with ‘fracture’ or ‘bone’ or ‘trauma surgery’ ‘orthopaedics’ or ‘orthopaedic surgery’. The terms were tested on the platforms to obtain the best search strategy. The following search criteria were applied: full-text-accessible articles, articles in English, peer-reviewed original research papers or relevant systematic reviews, without restriction to the year of publication. Additional literature was sourced by reviewing the reference lists of all the studies identified from the database search.

### 2.2. Literature Selection and Analysis

Two researchers (C.K.U. and N.U.) independently assessed the search results and screened the studies for relevance. The eligibility of the studies was evaluated in a stepwise approach: screening by title, followed by a critical reading of the abstracts and the full texts of potentially relevant studies. Differences between the two independent researchers were settled by a joint discussion with the other authors. We included only studies that specifically focussed on the application of VR systems in orthopaedic trauma procedures and performed a narrative synthesis of the extracted data using a thematic approach. Reports describing VR systems primarily developed for elective arthroscopy or arthroplasty procedures were excluded from the synthesis and discussion, but those adaptable for trauma procedures are highlighted in the appropriate contexts. Each author independently re-evaluated the extracted data and the initial narrative synthesis during the manuscript preparation process and the final draft were critically reviewed by two orthopaedic surgeons experienced with VR tools.

## 3. VR Systems for Bone Trauma Procedures

### 3.1. Background

Based on technical features and capabilities, VR platforms may be loosely categorised into low- and high-fidelity systems. High-fidelity VR modalities enhance immersion by simulating clinical and surgical environments and procedures with increased interactivity, visual accuracy and appeal, and content specificity. They allow the replication of most or all aspects of a procedure or technique, closely emulating the operating room environment [24,38,39]. Conversely, low-fidelity VR systems replicate single or multiple tasks with restrictions on interactivity, visual presentation, available content, and commands. Such modalities permit the demonstration of specific aspects of procedures or techniques, enabling the rapid and repetitive simulation of a skill to attain proficiency, and are therefore suitable as a basic learning platform for junior trainees. Compared to low-fidelity modalities, high-fidelity systems are more expensive and require more complex logistical considerations for set-up [24,38,40]. Most clinically available orthopaedic trauma VR systems fall into the high-fidelity category. Patient-specific VR simulations are enabled by the incorporation and conversion of MRI or CT data into 3D models using image processing and visualisation software. Examples of such software for quantitative analysis and visualisation of medical images include MIPAV (Medical Image Processing, Analysis, and Visualization) [41], Medical Imaging Interaction Toolkit (MITK) [42], 3D Slicer [43], Simpleware [44], 3D-DOCTOR [45], and Osirix [46].

While there is currently no formal classification system for VR modalities in orthopaedic trauma, for convenience, we have categorised existing simulation modalities into those for fracture fixation techniques, drilling procedures, and prosthetic design and implantation. Table 2 summarises the essential features of the described categories of orthopaedic trauma VR systems. It should be understood, however, that some multimodal simulators encompass diverse procedural modules and may not technically be restricted to a specific category.

### 3.2. Fracture Fixation VR Simulators

Surgical repair of bone fractures involves the reduction and fixation of the fracture fragments by means of screws, plates, or implants as appropriate for the nature and complexity of the fracture. Procedural accuracy is imperative for creating optimal mechanical and biological conditions for bone healing and restoration of bone anatomy and function. In addition, when an articulating joint like the hip, knee, or ankle is affected, it is essential to repair the bone joint surface precisely to prevent post-traumatic osteoarthritis. Enhanced surgical precision in orthopaedic trauma improves procedural safety, promotes proper anatomic healing, and reduces post-operative morbidity and long-term complications such as delayed, mal- or non-union, and post-traumatic osteoarthritis [38,47,48,49]. Five procedural phases of preoperative VR planning of fracture fixation have been described, namely, the generation of patient-specific geometrical models, fracture reduction, fixation, analysis of surgical planning, and intra-operative guidance [50]. A number of experimental and clinically applicable VR systems have been developed to accomplish one or more of the fracture fixation procedure phases.

The TraumaVision software developed by Swemac, Melerit, and Simulution Inc. (Burnsville, MN, USA) simulates a variety of orthopaedic trauma scenarios, including femoral neck fracture, trochanteric fracture, subtrochanteric fracture, femoral shaft fracture, pelvis fracture, spinal surgery, slipped capital femoral epiphysis, and Motec^®^ wrist prosthesis [51,52]. The software contains several preloaded training modules such as drill skills, cannulated screws, dynamic hip screws, and fluoroscopy training. A conventional A-P and lateral radiograph records fluoroscopy, which is provided by pushing a foot-controlled paddle. Phantom Omni, a computer-connected robot arm controllable with either of the operator’s hands, mimics operating tools and provides haptic feedback [52]. A Geomagic Touch X (Geomagic, Cary, NC, USA) haptic device may also be employed for positional sensing and precise force-feedback output, enabling the operator to appreciate tissue and bone resistance and even differentiate cortical and cancellous bone [53]. During a simulated operation, the software tracks a variety of performance parameters and reliably discriminates between novices and experts [52].

The BoneDoc DHS simulator is a web-browser-based non-haptic VR platform that operates on a regular PC and mouse and simulates screw and plate fixation of hip fractures [54]. Two-dimensional radiographic images facilitate fracture reduction and 3D implant placement on a virtual hip model. The simulated decision-making steps in the program include placing the C-arm to examine the fracture, reduction of the fracture using the virtual fracture table, skin incision, determination of entry location, guide wire angulation and depth of entry, and installation of the lag screw and cortical screws supporting the side plate [54,55]. The simulator also includes an assessment feedback component, and despite the lack of haptic function, the simulator was shown to have good face validity (closely mimics the actual procedure) [54]. Besides the lack of haptic/somatosensory feedback, the other remarkable limitation of this method is the absence of the psychomotor movements performed in surgery.

The first patient-specific biomechanical model developed by Boudissa et al. could replicate success or failure in the virtual reduction of an acetabular fracture depending on the selected surgical reduction strategy and sequence [56]. To recreate intraoperative bone fragment behaviour and reduction quality, the model simulated the impact of forces on the fragments and soft tissue–bone interactions. The model demonstrated clinical feasibility and favourable intraoperative outcomes [57]. Buschbaum et al. developed and evaluated a VR system based on preoperative CT scans for the automated repositioning and planning of the optimal reduction approach for femoral fractures [58,59]. Using a reference-coordinate system for calculating reduction parameters allowed for effective planning of reduction approaches. Adjustments to the reduction parameters are applied progressively until the fracture target position is attained.

The SQ Pelvis software, a PC-based preoperative planning VR program with 3D visualisation and surgical simulation tools for pelvic and acetabular fracture, was developed by clinicians at the University Clinical Centre, Ljubljana, in collaboration with computer engineers from Sekvenca Inc. [60]. The program provides a virtual comprehensive procedural simulation, including osteosynthesis and C-arm simulations, based on actual patient information (patient CT data is in DICOM format). The software permits the movement and rotation of bone fragments in all three planes to enable reduction and subsequent fixation. In addition, it allows for selecting the appropriate reconstruction plate, automatic contouring of the plate to the reduced pelvis, control of the direction and length of screws, and comprehensive C-arm imaging during surgery. All procedure steps are recorded in a printable format for surgical documentation, patient education, or research. Preliminary studies of the SQ Pelvis software in clinical settings demonstrated good practical utility for preoperative planning and facilitation of actual surgical experience, and it is currently routinely used at the traumatology department of the University Medical Centre in Ljubljana.

A system that employs a novel blend of cutting-edge 2D/3D image processing and surface processing algorithms to virtually recreate shattered bone pieces for severity classification or preoperative planning in complex bone fractures was proposed by Liu et al. [61]. A number of cutting-edge algorithms for identifying, extracting, and reassembling virtual pieces are integrated into a single system to facilitate reconstruction. To aid surgeons in classifying the clinical severity of comminuted bone fractures, the approach allows for the extraction of quantitative information not previously accessible from fracture cases in terms of bone fragment data. Tibial plafond fractures, which are difficult-to-treat, complex fractures typically resulting from high-energy trauma such as a gunshot wound or road traffic accidents, were used as a model for the system. Similarly, Fürnstahl et al. developed a semiautomated virtual environment to reconstruct complex proximal humerus fractures. A contralateral matching algorithm showed the efficacy of contralateral bone modelling for this type of fracture and permitted precise and efficient fragment alignment. On cadaver specimens, the fracture reduction approach was associated with decreased procedural time and minimal translational displacement and rotational errors in the reconstructed bone geometries [62].

A volume-based orthopaedic surgery simulator for complex orthopaedic surgeries, including arthroplasty, corrective or open osteotomy, open reduction of fractures and amputation, was developed by Tsai et al. [63]. The system comprises multiple modules, including an interface module, a volume conversion module, an isosurface reconstruction module, a rendering module, and a simulation module. The software can simulate different orthopaedic procedures and provide stereographic images of the simulated geometric and topologic alterations to bones, prostheses, and bone grafts and is adaptable for both preoperative verification/rehearsal of surgical modality and surgical training.

The innovative minimally invasive plate osteosynthesis (MIPO) approach in orthopaedic trauma surgery requires substantial training for adequate skill acquisition. A VR platform was developed by Negrillo-Cárdenas et al. for training in the MIPO technique for surgical reduction of humeral supracondylar fractures [64]. It was shown that the common malrotation of MIPO-treated fractures might be avoided by enhancing the motor skills and expertise of surgeons via an emphasis on the mobility of bone fragments. By using high-quality lighting, post-processing effects, and comprehensive medical assets, the planned setting provides a genuine experience throughout the procedure.

VR systems have also been developed for procedures in non-appendicular bones. A surgical planning VR system for the evaluation and reconstruction of severe atrophic mandibular fractures was described by Castro-Núñez et al. [65]. The tool enabled mirror imaging, facilitating the alignment of pieces and the restoration of incomplete segments, and was found to be clinically beneficial in reducing surgical time and delivering predictable outcomes. Rambani et al. developed a desktop-based simulation for the training of pedicle screw insertion in the lumbar spine and a Computer-Assisted Orthopaedic Training System for fracture fixation with Polaris optical tracking-based haptic functions and demonstrated a significant improvement in procedural time, accuracy, and the number of exposures after the training on the simulator systems [66,67].

### 3.3. Orthopaedic Drilling Simulators

Precision drilling is an integral skill in orthopaedic surgery. Nonetheless, bone drilling is a delicate procedure that demands a high degree of dexterity and expertise. The associated high drilling resistance and intense vibrations make it difficult to grip the handpiece and escalate the risk of damage to the drill. An assessment of the impact of haptic feedback in VR simulation of cortical bone drilling using changes in drill plunge depth as an evaluative metric suggests that bone drilling simulation with haptic feedback is effective in simulating the required motor skill dexterity and control and leads to a decrease in soft tissue injury [68].

To simulate and predict the hip drilling process, Tsai et al. developed a volume-based surgical simulator with haptic functions utilising a force and torque computation model and demonstrated its practical application in screw and plate surgery for hip trochanter fracture positioning [69]. The force and torque computation model is employed to simulate haptic reactions in the drilling procedure based on patient-specific volumetric data harnessed to simulate the dynamics of the bone geometry during the surgical process. The calculated torques are used to evaluate the required work for drilling, the drill bend, and the hand-piece oscillation during the drilling process [69].

Vankipuram et al. developed a realistic drilling software deploying a Geomagic Touch X haptic device for visiohaptic interaction with virtual bones. The program permits horizontal drilling into the femoral body with simulated targets for precision and tracks and analyses the surgeon’s movements to assess surgical proficiency. The simulator was confirmed to have a learning effect that is transferable to actual drilling, and its surgical performance indicators could distinguish the hierarchy of expertise from senior surgeons, residents, and medical students [70]. Similarly, Sang-Won Han et al. demonstrated the clinical and educational effects of a VR simulation system designed to perform a high tibial osteotomy [71]. The system uses a morphable haptic controller that provides geometric and tactile feedback facilitating interaction between the surgical instruments and surgical sites.

The Haptic Orthopaedic Training (HOOT) simulator developed at the Imperial College London focuses on dynamic hip screw (DHS) operations for certain hip fractures [72]. Drilling a pilot hole through the outside border of the femur into the femoral head is a critical initial step in DHS surgery that determines its trajectory and ultimate outcome. However, precise positioning of the pilot hole is difficult to achieve since the surgeon cannot visually monitor the location of the guide wire’s tip and must rely on several X-ray images to track its progress (with obvious problems including repeated radiation exposures and extended procedure time). Accordingly, the HOOT project aimed to develop and validate a haptically enabled simulator for training guide wire placement in DHS surgery. Unlike other haptic applications, the HOOT simulator is unique in its use of W5D from Entact Robotics to produce both linear forces (x,y,z) and rotational forces (torques: roll, pitch, yaw) [73]. The researchers plan to enhance the realism of the haptic feedback and perform validation studies in the next phase of the project.

Pettersson et al. developed a surgical simulator for the drilling procedure in cervical hip fracture surgery. The volumetric dataset obtained from the patient’s CT scan is used to produce visuohaptic feedback by replicating fluoroscopic images and the drilling procedure. Prior to simulation, the bone must be relocated into the correct position. An automatic segmentation based on nonrigid registration with the Morphon method is deployed to identify the fracture’s constituent pieces and estimate the link between the fracture elements [74].

### 3.4. Prosthesis Development and Implantation VR Modalities

This category of VR systems is mostly applied to elective orthopaedic procedures but is nevertheless highlighted for potential relevance in certain contexts related to orthopaedic trauma.

The Virtual Operation Planning in Orthopaedic Surgery (VIRTOPS) software is a VR system for 3D planning and simulation of hip and pelvic surgeries, including endoprosthetic hip reconstruction with hemipelvic replacement [75]. The software also facilitates the personalised design of anatomically flexible, modular prostheses for bone tumour surgery [76]. A patient-specific 3D hip model is created from CT images, and an ROI-based segmentation separates the bone tumour in multispectral MRI sequences; CT and MRI data are then fused by a segmentation-based registration approach enabling visualisation of the tumour position. Texture mapping, quantitative parameter colour coding, and transparency aid in optimal prosthesis positioning and geometry. Virtual models can interact in 3D using stereoscopic visualisation tools and 3D input devices. The virtual planning environment eliminates the need for expensive solid 3D models and permits the comparison of different surgical approaches. The generated 3D images and videos can be utilised for patient preoperative educational counselling and surgical planning documentation [75,76].

Another prosthetic placement VR modality is HipNav, an image-guided surgical navigation system. It combines a 3D preoperative planner, a simulator, and an intraoperative surgical navigator to precisely measure and direct the placement of prosthetic components during total hip arthroplasty [77]. The use of virtual surgical planning to fabricate the required hardware in preparation for open reduction and internal fixation of atrophic edentulous mandible fractures was reported by Maloney et al. [78]. As already noted, multimodal VR systems such as the SQ Pelvis [60] and TraumaVision software [51,52] also consist of prosthetic design and implantation modules.

**Table 2 medicina-59-00562-t002:** Summary of bone trauma virtual reality (VR) systems.

Category	Examples	Procedures	Remarkable Features
Fracture fixation/ Bone drilling/prosthesis design	TraumaVision software [51,52].	Femoral, pelvic, and wrist fracture repair	Preloaded training modules such as drill skills, cannulated screws, dynamic hip screws, and fluoroscopy training
Fracture fixation	BoneDoc Dynamic hip screw (DHS) simulator [54,55].	Screw and plate fixation of hip fractures	Simulates several procedural decision steps and enables feedback assessment; non-haptic
Fracture fixation	Boudissa et al. [56].	Virtual reduction of an acetabular fracture	Patient-specific biomechanical model
Fracture fixation	Buschbaum et al. [58,59].	Reduction of femoral fracture	Uses a reference-coordinate system for the calculation of reduction parameters
Fracture fixation/prosthesis design	SQ Pelvis software [60].	Pelvic and acetabular fracture repair	Comprehensive procedural simulation, including osteosynthesis and C-arm simulations
Fracture fixation	Liu et al. [61].	Complex fracture repair	Preoperative planning and severity classification
Fracture fixation	Fürnstahl et al. [62].	Reconstruct complex proximal humerus fractures	Semiautomated virtual environment; contralateral bone modelling
Fracture fixation	Tsai et al. [63].	Open reduction of fractures and other complex surgeries like arthroplasty, corrective or open osteotomy, and amputation	Volume-based orthopaedic simulator
Fracture fixation	Negrillo-Cárdenas et al. [64].	Minimally invasive plate osteosynthesis (MIPO) technique for surgical reduction of humeral supracondylar fractures	Uses high-quality lighting, post-processing effects, and comprehensive medical assets
Fracture fixation	Castro-Núñez et al. [65].	Evaluation and reconstruction of severe atrophic mandibular fractures	
Fracture fixation	Rambani et al. [66,67].	Pedicle screw insertion in the lumbar spine	Polaris optical tracking-based haptic functions
Bone drilling	Tsai et al. [69].	Screw and plate surgery for hip trochanter fracture positioning	Uses a force and torque computation model
Bone drilling	Vankipuram et al. [70].	Horizontal drilling into the femoral body	Geomagic Touch X haptic device
Bone drilling	Sang-Won Han et al. [71].	High tibial osteotomy	Uses a morphable haptic controller that provides geometric and tactile feedback
Bone drilling	Haptic Orthopaedic Training (HOOT) simulator [72].	Dynamic hip screw (DHS) operations	For training guide wire placement in DHS surgery, uses W5D from Entact Robotics to produce both linear forces and rotational forces.
Bone drilling	Pettersson et al. [74].	Cervical hip fracture surgery	Automatic segmentation based on nonrigid registration with the Morphon method
Prosthesis design/implantation	VIRTOPS [75].	Endoprosthetic hip reconstruction with hemipelvic replacement	Facilitates the personalised design of anatomically flexible, modular prostheses
Prosthesis design/implantation	HipNav [77].	Placement of prosthetic components during total hip arthroplasty	Combines a 3D preoperative planner, a simulator, and an intraoperative surgical navigator
Prosthesis design/implantation	Maloney et al. [78].	Preparation for open reduction and internal fixation of atrophic edentulous mandible fractures	

## 4. Clinical Translation and Efficacy of Orthopaedic Trauma VR Modalities

Several studies have reported the advantages of both high- and low-fidelity VR over conventional surgical training modalities [34,79,80,81,82,83,84,85]. In the current era of surgical training, it has become increasingly clear that the required skill sets in specialist orthopaedics training cannot be solely realised in the clinical setting. Indeed, in recent decades, practical teaching and learning outside the operating room have become a compelling necessity in orthopaedic surgery, mostly due to rising training costs and ethical concerns over traditional pedagogical approaches [32,33]. VR simulation systems are well-adapted to meet the current training gaps in orthopaedic residency programmes. Table 3 summarises the potential advantages of VR-based modalities in orthopaedic trauma surgery. In addition, different studies have also assessed the reliability and clinical validity of existing VR systems applicable to orthopaedic trauma procedures. Reliability refers to the repeatability and accuracy of the modality, while validity assesses if the simulator is teaching or assessing the intended objective [86]. Different concepts of validity are applied in assessing the validity of VR simulation systems. Face validity evaluates how realistic a simulator is; content validity evaluates how appropriate it is as a training tool; concurrent validity measures how well it corresponds with the gold standard; predictive validity assesses how well it forecasts future performance; construct validity measures its ability to differentiate between a skilled and unskilled user; and transfer validity assesses whether training on the simulator results in skill transfer to a realistic environment [86,87].

Validity assessments of bone trauma VR simulators show wide methodological variability, and no formal evaluation protocol currently exists. Twenty-two orthopaedic surgery residents participated in a stratified, randomised, controlled study that evaluated their performance on a virtual haptic-enabled ulnar fracture fixation simulator compared to the conventional synthetic manikin simulator (Sawbones simulator). It was noted that both methods demonstrated evidence of construct validity (capacity to evaluate trainees’ technical skills), but although the virtual simulator showed potential value as a surgical educational tool, it failed to attain similar standards as the Sawbones simulator [88,89]. Compared to a conventional learning method, an immersive VR system was shown to efficiently teach a complex surgical procedure (optimum glenoid exposure in shoulder arthroplasty) and demonstrated face, content, construct, and transfer validity [90].

When compared with a technique guide, VR was shown to improve both the procedural accuracy and completion rate in intramedullary tibial nail insertion in a recent randomised control trial, demonstrating its capacity to assist trainees in understanding surgical processes and manoeuvres [91]. A similar randomised control study found that training novice medical students to perform a simulated tibial intramedullary tibial nail operation using an interactive VR simulation had superior results than training with a passive standard guide [92]. Clinically relevant objective performance indicators of inexperienced surgical trainees, including total procedural time, fluoroscopy time, number of radiographs, tip–apex distance, attempts at guide wire insertion, and probability of cut-out, were significantly improved following exposure to a VR DHS simulation with the TraumaVision (SveMac, Sweden) haptic- and fluoroscopy-enabled VR simulator [93].

A recent randomised, controlled, double-blinded trial assessed the educational value of integrated haptic feedback in a VR bone drilling simulation comparing the performance (assessed by plunge gap distance, drilling time, and objective structured assessment of technical skills) of 31 junior surgeons randomly allocated to a haptic or non-haptic group. The trainee doctors experienced a VR training module for drilling bicortical holes for screw insertion on a VR tibia bone model with either haptic or no haptic feedback, followed by an ex vivo identical test on a tibial sawbone model. The authors proved the higher performance of participants included in the haptic group, highlighting the potential educational role of haptic feedback in orthopaedic training simulation models [94].

For training in the distal interlocking of intramedullary nails, the Digitally Enhanced Hands-On Surgical Training (DEHST) concept and technology scored highly in terms of training capability and realism and reliably differentiated between experts and novices [95]. DEHST is a novel modular surgical skills training and evaluation system incorporating digital tools with haptic stations for hands-on learning. Position and orientation may be tracked in six directions from a single plane of the projected image using a specialised optical tracking technique [96].

In a slipped capital femoral epiphysis model, orthopaedic trainees rated VR training as more valuable than reading/video methods despite similar performance outcomes (surgical time, Global Rating Scale score, radiographic or physical accuracy of screw position, or articular surface breaching) to training with physical simulations [97]. Following training with immersive VR to revise a failed percutaneous pinning of a slipped capital femoral epiphysis, Lohre et al. reported an immediate skill acquisition and transfer to a real operating setting by a senior orthopaedic trainee [98]. Similarly, superior learning efficiency, knowledge acquisition, and skill transfer were found among senior orthopaedic surgery residents who underwent surgical training with an immersive VR training platform offering a case-based module for reverse shoulder arthroplasty for advanced rotator cuff tear arthropathy [99].

Virtual preoperative planning enables the precise evaluation of fracture features and simulation of fracture reduction and internal fixation prior to surgery. Several studies have also evaluated surgical outcomes with virtual preoperative planning compared to traditional preprocedural planning. A range of intraoperative and postoperative indices have been applied for comparison, including surgical duration, intraoperative blood loss, radiation exposure frequency, clinical and radiographic quality of fracture reduction, and postoperative complications. Overall, compared to conventional planning, VR-based preoperative planning for fracture fixation was shown to be associated with enhanced intra-operative efficiency (decreased operating length, bleeding, and fluoroscopy frequency, and better fracture reduction quality) and superior postoperative outcomes (improved functional outcomes, quicker fracture healing, fewer postoperative morbidities, hospital readmissions, and reoperations) [48,50,57,100,101,102,103,104,105].

Immersive VR has also been exploited as a form of distraction therapy and adjunct to surgical anaesthesia to alleviate pain and anxiety and promote patient relaxation [106,107,108]. However, Peuchot et al. reported that the use of VR for immersive distraction during total knee arthroplasty under spinal anaesthesia did not seem to reduce patient anxiety or alter patient satisfaction, although improvements in postoperative comfort and reductions in intraoperative complications were achieved [109]. In addition, as a rehabilitation tool, VR training, compared to sensory–motor training, was found to have a more favourable impact on pain, functional disability, and the modification of inflammatory biomarkers in post-traumatic osteoarthritis after an anterior cruciate ligament injury, although a negligible effect on bone morphogenic proteins expression was reported [110].

## 5. Drawbacks of VR in Bone Trauma

While several advantages and benefits of VR simulation systems in bone trauma surgery have been highlighted, some important drawbacks and limitations should be equally understood. Currently, most immersive VR technologies suffer remarkable limitations with regard to image quality, degree of immersion, haptic accuracy, and other technical issues (e.g., battery life and wireless technologies) [111]. The distressing phenomenon of cybersickness (signs and symptoms related to VR experience) is another important drawback of immersive VR systems [112]. Many of the available bone trauma VR systems lack the requisite computing power and high-performance software architecture to provide optimally reliable audio–visual and haptic output. Further developments in haptic technology are specifically required to overcome the current limitations of haptic devices for VR simulation of fracture procedures, including challenges in a high-fidelity virtual recreation of sensations of density and palpability of convex surfaces, the limitations in maximum force, multi–modal telepresence, and teleaction in conventional haptic devices, and the high-bandwidth requirement barrier for an online haptic system [31,113]. In addition, a wide variability currently exists among the different VR tools in terms of technical quality and functional value [99]. This is partly due to the lack of regulatory standards and metrics in the design and manufacture of clinical VR systems and the lack of uniform protocols for clinical validation. Furthermore, although VR technology represents a well-adapted option to meet the current needs of orthopaedic training programmes, few studies have examined the long-term implications of orthopaedic simulation training on the retention of new skills [114]. Meanwhile, cost remains a prohibitive barrier for high-fidelity bone trauma VR systems, especially in resource-limited settings. Nevertheless, future VR tools are expected to be cheaper, more user-friendly, and in tandem with technological advancements [38,109].

## 6. Conclusions and Future Perspectives

Over the past two decades, an increasing number of VR systems for orthopaedic trauma surgeries have been introduced, although a considerable number are still in the experimental phases. The use of VR simulation modalities minimises the reliance on patients, cadavers, and animals for surgical training and skill practice. VR training simulators aid in developing a visuospatial appreciation of anatomy and provide learners with a controlled, low-risk environment to practise techniques before attempting them on a real patient [31]. Besides facilitated procedural learning and enhanced preoperative planning, VR systems have been shown to improve a broad range of intraoperative and postoperative outcomes. Nevertheless, clinical outcomes may be further enhanced by including biomechanical analysis within the framework of VR preoperative planning for fracture repair [50]. Rigorous validation of existing orthopaedic trauma VR systems in relation to the face, content, concurrent, and transfer validity is crucial, in addition to further development of orthopaedic VR systems in line with industry standards and metrics of immersion, multisensory realism, and versatility. The design and reporting guidelines proposed by the European Association of Endoscopic Surgeons Work Group for Evaluation and Implementation of Simulators and Skills Training Programmes represents an important step for harmonizing and standardising the appraisal of simulation studies [115]. Additional research is also warranted to demonstrate the cognitive simulation validity of immersive VR platforms, skill retention and decay with VR, translation to patient-derived outcomes, as well as the security, privacy, and cost-effectiveness of current orthopaedic trauma VR systems [38,114].

## Figures and Tables

**Figure 1 medicina-59-00562-f001:**
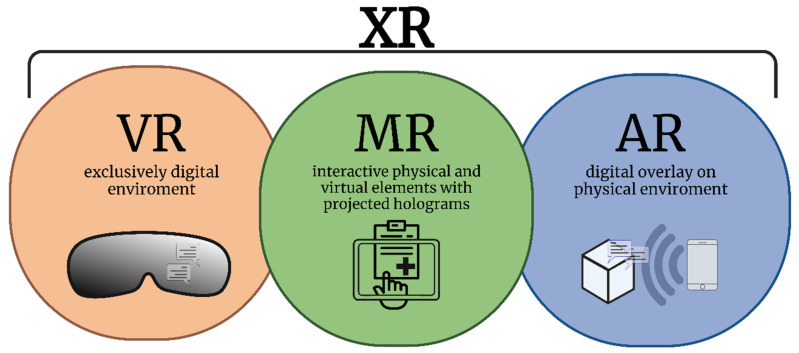
The spectrum of surgical reality systems. VR = virtual reality; MR = mixed reality; AR = augmented reality; and XR = extended reality (umbrella term encompassing all three modalities).

**Table 1 medicina-59-00562-t001:** Essential features of virtual, augmented, and mixed reality surgical systems [29,31].

Virtual Reality	Augmented Reality	Mixed Reality
Complete visual immersion in an artificial digital (computer-generated) environment. Uses 2D/3D graphic software for reconstructing clinical CT/MRI images. Visual experience may be complemented by incorporating haptic devices, artificial sounds, and other stimuli. Current hardware support systems include the Touch 3D Systems, W5D Enact Robotics, Ascension trakSTAR, Optotrak optical tracking camera, smartphones, tablets (e.g., Google Cardboard or Samsung Gear VR), mice, etc. Suitable for preoperative planning/rehearsal, patient education, and surgical training/assessment.	Digital images superimposed on real-world physical surfaces. Uses 3D graphic software for reconstructing clinical CT/MRI images. Enhances the operator’s perception of depth. Current hardware supports systems include smartphones, tablets, and both optical and video see-through-based head-mounted displays such as the Google Glass or Microsoft HoloLens. Possibility of remote training experience. Suitable for pre-op planning, intra-op surgical navigation, and surgical training/assessment.	Most functionally advanced surgical reality technology combining an interactive digital display overlay and projected holograms. Uses 3D graphic software for reconstructing clinical CT/MRI images. Requires less preoperative calibration and allows a greater degree of freedom for preoperative image reconstruction. Better control of intraoperative visualisation and the possibility of remote communication, e.g., via Skype. The most adapted hardware support system currently is the Microsoft HoloLens. Suitable for pre-op planning, intra-op surgical navigation, and surgical training/assessment.

**Table 3 medicina-59-00562-t003:** Potential benefits of bone trauma virtual reality systems.

Enhanced procedural teaching and learning efficiency
Decreased use of animal models, cadavers, and patients for procedural teaching and practice Avoidance of ethical issues associated with traditional procedural teaching and learning Faster knowledge acquisition and visuospatial appreciation of surgical anatomy Improved psychomotor skill acquisition and appreciation of complex surgical manoeuvres Better knowledge and skill retention and clinical transfer Improved procedural accuracy Reduced number of procedural learning attempts Increased procedural completion rate Objective assessment of surgical proficiency and tracking of learning progression
Enhanced preoperative planning
Enhanced preoperative patient counselling and education Low-risk environment for preoperative procedural rehearsal/practice Objective and comprehensive procedural documentation Objective analysis and evaluation of different surgical approaches Objective prediction of intraoperative complications Design and analysis of patient-specific prosthetic models
Enhanced intraoperative efficiency
Decreased procedural time (intraoperative duration) Decreased radiation exposure (e.g., fluoroscopy frequency) Decreased intraoperative blood loss Reduced surgical invasiveness Improved clinical and radiographic quality of fracture reduction
Improved postoperative outcomes
Decreased rehabilitation and fracture healing time Improved return of functional capacity Decreased postoperative complications Decreased incidence of hospital readmissions Decreased fracture repair revisions

## Data Availability

Not applicable.

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
