# Peer review of "Application of Virtual Reality Systems in Bone Trauma Procedures"

_medicina, 2023, doi:10.3390/medicina59030562_

Round 1

Reviewer 1 Report

The authors presented a detailed review of VR systems for orthopaedic trauma procedures, however, I have a few concerns about the study;

-         The excessive details such as in the introduction part of the VR systems, resulted in the loss of integrity of the manuscript. Briefly designed review studies have a higher scientific impact since they ensure readers focus on the topic.

- The study aimed to reveal the state of the VR systems for orthopaedic trauma procedures however none of the authors was an orthopaedic surgeon. Furthermore, there was no discussion revealing the VR systems for trauma procedures from the orthopaedic surgeons’ point of view.

Reviewer 2 Report

Dear authors,

Thank you for the great effort on this manuscript I found it very interesting.

However, there is a need to revise the manuscript according to the narrative review format. A specific topic on "methodology" should be included where indicating the sources of information used. Those sources of information if derived from multiple databases, shall be described in the selection of articles. They shall be analyzed and synthesized, and how was internal and external peer review conducted and used in the finalization of the manuscript.

Furthermore, the background and the findings from the narrative review shall be arranged separately.

Ultimately, the abstract shall be refined, for example, to list out all the databases

Above all, the information gathered was comprehensive and interesting.

Reviewer 3 Report

A good article which explain the novelty in future medicine and traumatology. 
In the future I would like to see more applications in other Orthopaedic fields.
